# Developing a socio-ecological model for community engagement in a health programme in an underserved urban area

Lizzie Caperon[1]*, Fiona Saville[2], Sara Ahern[1]

1 Bradford Institute for Health Research, Bradford, United Kingdom, 2 Better Start Bradford, Mayfield Centre, Bradford, United Kingdom

* elvcaperon@yahoo.co.uk

## Abstract

Despite a recent increase in community engagement in health initiatives during the COVID-19 pandemic, health inequalities and health inequities remain a serious problem for society, often affecting those in underserved communities the most. Often individualised incentives such as payment for vaccinations have been used to increase involvement in health initiatives but evidence suggests that these do not always work and can be ineffective. This paper addresses the real world problem of a lack of involvement of communities in health programmes and subsequent health inequalities. Using data from nine workshops with community members evaluating a large community health programme, we develop a socio-ecological model [SEM] of influences on community engagement in health programmes to identify holistic and systemic barriers and enablers to such engagement. To date SEM has not been used to develop solutions to improve community engagement in health programmes. Such an approach holds the potential to look beyond individualised conceptualisations of behaviour and instead consider a multitude of social and cultural influences. This knowledge can then be used to develop multi-faceted and multi-layered solutions to tackle the barriers to community engagement in health programmes. Our SEM highlights the overarching importance of the socio-cultural environment in influencing community engagement. Within the socio-cultural environment were factors such as trust, social support and community mindedness. We also found that other factors affecting community engagement fall within individual, economic, technological, political and physical environments. Such factors include engagement in community organisation governance and processes, access to and ability to use technology and access to safe outdoor spaces. We propose further testing our socioecological model in other communities.

## Introduction

The COVID-19 pandemic has led to a fundamental shift in approaches to community health [1]. There has been an increase in participation in health prevention, awareness raising and community-led health-based activities during COVID-19 with estimates of a million people

**Data Availability Statement:** All relevant data are within the manuscript and its Supporting Information files.

**Funding:** Bradford Institute for Health Research received funding for this service design process the

Big Lottery Fund, UK (https://www.tnlcommunityfund.org.uk/) as part of the A Better Start programme. The funders had no role in study design, data collection and analysis, decision to publish, or preparation of the manuscript.

**Competing interests:** The authors have declared that no competing interests exist.

volunteering to support the pandemic response in the UK [2] and mutual aid groups providing support to citizens in countries throughout the world [3]. Global health guidelines and other research has highlighted that such community participation in health is crucial [4–6], especially the inclusion of perspectives from diverse and harder to reach communities [7]. However, despite a recent increase in community engagement in health initiatives, health inequity and inequalities persist [8, 9]. The pandemic has uncovered amplified systemic health and socioeconomic inequities, especially in the geographical area in West Yorkshire in which our study takes place [10]. The context of the COVID-19 pandemic has amplified the need to consider effective ways to improve community engagement in health programmes. During the COVID-19 pandemic, for example, vaccine hesitancy, particularly in underserved communities [6, 11–14], has highlighted that resistance to participation in health initiatives is a significant problem, as it was before COVID-19 [15]. Common attempts to address health problems in community settings have involved individual incentives [16, 17]; payment or compensation for receiving vaccine doses being some of the most recent and prevalent examples, which research has found has variable effects and in many cases does not work [18, 19]. In contrast, broader community-led solutions are being increasingly seen as key to tackling health inequalities [9], and research has pointed to the importance of community-level factors in influencing health outcomes [20].

Socioecological systems theory acknowledges the central influence of beliefs and ideologies in society whilst accounting for the interconnections amongst individuals and uses a holistic socio-ecological lens with which to understand behaviour [21]. The first socio-ecological model [SEM] was first introduced to understand human development in the 1970s and was formalised as theory in the 1980s [22]. Since SEM was conceived in the 1970s, there have been many interpretations of SEM models to develop multilevel approaches to areas such as public health promotion, violence prevention, healthy college campuses, safe practice in primary care and bowel cancer prevention to name a few [23–26]. Furthermore, the Centres for Diseases and Prevention in the US who have adapted the SEM for health promotion efforts to include spheres of organisational, community and policy [27]. Therefore, adopting a socioecological lens to understand and explore community engagement in community health initiatives brings with it great potential to develop multi-level solutions which cross multiple influential environments and tap into the social determinants of health which so often drive people's decision making and therefore tackle health inequalities [28, 29]. Taking a socioecological approach allows us to improve interventions across multiple system levels which lead to solutions not only within the health system but also through across political, physical, socio-cultural and other structures in society. Socio-ecological models hold great value in considering the interaction of behaviours across multiple levels of influence and lead to multi-level suggestions for interventions to effectively influence behaviour [27, 30–32]. Some socio-ecological models see cultural context as important in interventions [33, 34]. The complex role played by context in the development of health problems is connected to the social [28, 29] and structural determinants of health [9].

Though socioecological models have been developed to consider influences of lifestyle behaviours such as dietary behaviour [32, 35] and on physical activity [36] and whilst one recent study looked at how SEM could be used to engage diverse populations in a research programme [37], to the best of our knowledge there has been no exploration to date of socioecological influences on community engagement in health programmes. The purpose of this paper is to develop a socioecological model of behavioural influences on community engagement in a community health programme. Our paper therefore addresses the real world problem of a lack of involvement of communities in health programmes and subsequent health inequalities. We also address the gap in the current knowledge by developing a socio-ecological model

[SEM] of influences on community engagement in health programmes to identify systemic barriers and enablers to such engagement. Such an understanding holds the potential to look beyond individualised conceptualisations of behaviour and consider a multitude of social and cultural influences. This knowledge can then be used to develop multi-faceted and multi-layered solutions to tackle the barriers to community engagement in health programmes.

## Setting

Community engagement has been found to be particularly effective in public health interventions for disadvantaged groups [38]. Our study takes place with a community in an economically underserved urban area within three electoral wards in which the Better Start Bradford programme operates in Bradford, in northern England. Bradford was hit particularly hard by the COVID-19 pandemic [10]. The wards BSB serves are considered amongst the most underserved in England and have populations made up of many different ethnicities [39]. Bowling and Barkerend is the one ward which BSB serves. It has a total population of 22,200, 11.2% of houses in the ward are overcrowded and 29.2% of the population are under 16. Life expectancy in the ward is lower than the district average [73.9 for men and 78.6 for women] and the ward is ranked 3rd out of 30 in the District for the 2019 Index of multiple deprivation. 42.7% of the population in the ward are white and 32.9% are Pakistani, with the remaining 24.3% made up of a range of other ethnicities. The dominant religion in the ward is Muslim [45.8%] with Christian [29.7] and no religion [14.6%] second and third respectively [40]. Bradford Moor is the second ward BSB serves with a population of 21,310 it is similar in size to Bowling and Barkerend but more houses [17.3%] are overcrowded in the ward. Like Bowling and Barkerend, life expectancy in the ward is lower than the district average [74 for men and 80 for women]. Bradford Moor is ranked 4 out of 30 wards in the District for the index of multiple deprivation. Demographically, 63.9% of the population of the ward are Pakistani and 17.3% are White with the remainder a mixture of different ethnicities. 72.8% of the ward are Muslim, 13% Christian and the remaining 14.2% belong to religions or have no religion [41]. Little Horton is the final district BSB serves with a population of 23,340 it is the largest of the three wards. 14.1% of homes are overcrowded and like the other two wards; life expectancy is below the district average at 76.3% for men and 82.2 for men. Little Horton is ranked 2nd of the 30 Wards in the District for the 2019 index of multiple deprivation. The majority of the population is Pakistani [48.5%] with 28.8% white. The main religion in the ward is Muslim [58%] with 24.7% Christian and the remaining 17.3% a mixture of other religions or with no religion [42].

The Better Start Bradford [BSB] Programme is funded by the National Lottery Community Fund over 10 years 2015–2025. The programme aims to improve the health outcomes of children by commissioning projects and services aimed at pregnant women and families with children under the age of 4. Engaging families with these projects and services, as well as with key programme messages promoting healthy child development, is therefore crucial to the success of the programme. The BSB programme includes community members in a range of governance roles and in engagement roles. The Family and Community Engagement [FACE] team are link workers employed by BSB to develop community partnerships, support parent led group activities and recruit volunteers to the BSB Programme Community champions are members of the community who promote the BSB programme activities. In October 2020 we set out to develop a logic model for community engagement in the BSB programme as part of the programme service design process. This co-production process involved running nine workshops with community members including pregnant women and parents of children under 4 which the BSB targets, FACE team members, health professionals, researchers and BSB staff. These workshops explored the barriers to the community engaging in the

programme as well as enablers which increased community engagement in the programme. As we were co-producing the logic model for community engagement we started the notice common themes on the influences of behaviour [barriers and enablers] for community engagement coming out of the workshops. As a result we decided to put these themes together into a socio-ecological model to more systematically and holistically explain why people do or do not engage in community health programmes. Therefore, this theoretical paper documents the barriers and enablers to community engagement we gathered from the initial service design process. This paper does not provide an evaluation of participation in the BSB programme; rather it explores the barriers and enablers to community engagement through a socio-ecological lens, using the BSB programme as a case study. The research team from the research and evaluation arm of BSB, the Better Start Bradford Innovation Hub, that is the evaluation arm of the BSB programme located within the Bradford Institute for Health Research, led the interpretation of findings documented in this paper.

## Methods

### Participants

Our participants had all been recruited to take part in the service design co-production process to evaluate the BSB Programme. These participants had been identified through discussion with members of the BSB team, specifically the Family and Community Engagement [FACE] team. Following their initial involvement in a coproduced service design process, participants consented to us using their data in follow up research, which this study represents. Participants comprised of 10 community members, volunteers [parents in the lead panel members, community champions and community board members] and other stakeholders [ward officers and social prescribers] who represented a range of ethnicities and genders and lived in the BSB target area to take part in our nine workshop sessions. Pregnant women and parents with children under the age of 4, who are the target population for the BSB programme, were included in the service design process and a range of participants offered different perspectives. Also included in the workshops were seven BSB team members including FACE team members. These were included to ensure that trusted community workers with whom other community members were familiar, were present. This also ensured that community members felt at ease and comfortable in the workshops.

### Recruitment and consent

FACE team members contacted all participants, either in person or by telephone and asked them if they would like to take part in service design process. Participants had the objectives of the service design explained to them by the BSB team and if they agreed to participate, written consent was obtained. All workshops were conducted online [according to COVID-19 restrictions] via ZOOM, a videoconferencing platform increasingly being used to conduct qualitative research [43]. Three community members attended all nine workshops, however many attended two or three of the workshops, and the composition of the group in the workshops changed every week. The total number of participants over all workshops was 25. To provide consistency for attendees, sessions took place at the same time weekly for 9 weeks. The service design process was part of ongoing community engagement activities which participants had volunteered to take part in as part of Better Start Bradford's ongoing community work. No direct quotations were used from the sessions in data analysis or write up, and no data provided would identify participants who were anonymised at the point of data analysis. All participants in the workshops provided written consent for their data to be used and for findings to be published.

## Workshop structure and guide

Workshops were initially thematically structured around the seven Scottish standards for community engagement [SSCE] which have been used in areas such as community planning and health and social care [44]. Workshops were conducted between November 2020 and March 2021. Nine workshops were conducted lasting 90 minutes each. In the first workshop the BSB research team discussed with participants that the sessions would aim to develop a community engagement strategy. Following the introductory workshop, seven subsequent workshops discussed each one of the SSCE in turn. The final workshop provided an opportunity for the research team to feedback findings and asked for feedback from community members. A workshop guide consisting of an agenda with questions that would be asked in the session was circulated prior to each workshop by the research team. Example workshop guides can be seen in S1 Table.

## Ethics

The process from which data in this article is derived was an exercise in service design which would go on to inform future service evaluation of community engagement in the Better Start Bradford programme. As such it did not require formal ethical review approval [HRA decision 60/88/81]. No identifying personal information was obtained or recorded for the purposes of research or any other use. All participants in the workshops provided written consent for their data to be used and for findings derived from the service design process to be published in this article.

## Analysis

Transcripts were taken from the workshops by two members of the research team. Following each workshop, the research team compared their transcriptions of the sessions to ensure accounts of the workshops were full and accurate. Framework analysis was used [45] and a matrix structure of key themes was developed based on the Scottish standards [see Table 1]. Data was analysed using above frameworks in Microsoft Excel and Nvivo. The data was then re-analysed along the socio-ecological themes we as researchers observed were emerging. Our themes were indexed systematically, a process which entailed comparison within and between themes. Our data analysis began with a priori codes from the socio-ecological model [SEM] literature. These codes were formed from the initial theory behind Brofenbrenner's first socio-

**Table 1. Levels of themes from the analysis.**

| Level of Influence | Theme | Sub-code [examples] |
|---|---|---|
| Individual | Individual environment | Time to take part in activates |
| Intermediate | Economic environment | Allocation of funding to local organisations for sustainable community engagement projects |
| | Technological environment | Development of a blended approach to community engagement using online and face-to-face methods [overlap with socio-cultural environment] |
| | Political environment | Improving transparency within the organisation to develop complaints procedures and disseminate organisational structure to show community where they fit into the organisation |
| | Physical environment | Reaching community members where they are. Door-knocking at different times of week for working people. [overlap with socio-cultural environment] |
| Higher/broader | Socio-cultural environment | Developing and sustaining strong relationships between community and community workers [BSB] built on trust. Increase capacity of BSB workforce to build and sustain these relationships with community. |

We present our results under each of these themes below.

ecological model which were nesting circles that placed the individual in the centre surrounded by various influential systems [22]. The microsystem closest to the individual contains the strongest influences and incorporates the interactions and relationships of the immediate surroundings. The outer rings of the models or outer environments have traditionally represented environments which have interactive forces on the individual such as community contexts or social networks [46]. SEM models illustrate that health behaviours are affected by the interaction between the characteristics of the individual, community and physical, social and political components.

Therefore our a priori codes drew on existing models, particularly models which place socio-cultural influences and community as an overarching macro influence on behaviours [35]. We began loosely therefore with three levels of codes–individual, intermediate and higher. However, as we began to code the data we discovered that specific environments were present within the intermediate and higher levels which were specific to the community engagement behaviour. Following this we began to formulate the dominant 'influences' or environments such as political, economic, physical and technological influences. Our analysis process was validated by discussing the analysis with the BSB team and research team.

## Results

Analysis of our data, both a priori and a posteriori, led to thematic representation under six environments; individual, political, physical, technological, economic and socio-cultural environments. Within each environment we coded specific aspects such as 'time to take part in activities' [individual] or 'allocation of funding' [economic]. We found that many codes overlapped several environments, as socio-ecological modelling encourages, showing multi-faceted influences on behaviour. The full list of codes and the environments these fit within can be found in S2 Table. Our list of levels of influence, themes and example codes can be found in Table 1.

### Individual environment influences on community engagement

Several individual factors were apparent in influencing community engagement. Examples of these included the individual's ability to use technology which in turn influenced whether they could take part in online community engagement activities [CEA's]. This factor overlaps with the socio-cultural, technological and economic influences on behaviour as often technological equipment and infrastructure were expensive for individuals to buy. If individuals did have access to these, sometimes their abilities to engage with the technological environment were limited by a lack of skills, time or technological awareness. Some community members also lacked the social support to learn or develop the abilities to use technology, and sometimes cultural or language barriers stood in the way of them accessing CEA's. Other individual factors included the ability [often connected to time available] to take part in CEA's or to be part of community engagement governance processes which could amplify their voices. This factor overlapped with the socio-cultural and political environments. Often social pressures, a lack of social support or cultural expectations led participants, particularly female community members, bearing the brunt of childcare responsibilities. This led to less time to engage with CEA's. Many community members stated they had problems finding time to engage in CEA's when juggling paid work, housework, childcare and other responsibilities. Furthermore, some community members lacked time to engage with BSB governance structures provided by the community action group and informal monthly meetings for community champions. This was connected with the support individual community members received in the form of training to allow them to take part in BSB governance structures and to gather and feed comments

from their communities back up through existing governance structures. Those individuals who had time had access to training and therefore involvement in the governance structures. The opposite was the case for those individuals who did not have time to engage.

## Economic environmental influences on community engagement

Several environmental influences affected community member's engagement with the BSB programme. These included the economic ability of the individual to buy or own technological equipment or the infrastructure [Wi-Fi, mobile data] which enabled their engagement in online activities necessary during the COVID-19 pandemic. Further factors affecting engagement were community members' perceptions that funding for the BSB programme was not being spent according to community needs, leading to reluctance to engage with the programme. Therefore community members wanted to ensure that funding was allocated to local organisations rather than to larger, less connected organisations that would leave the community when funding ended. Community members stated that if the funding processes within the BSB programme were more transparent and funding was made available to local initiatives, they would be more willing to participate in BSB activities.

## Technological influences on community engagement

Many community members stated a preference for the development of a blended approach using social media channels, local radio and TV with face-to-face engagement to ensure maximum accessibility for as many people in the community as possible. Another technological influence was the provision of a range of new, accessible methods to improve engagement including, for example, the introduction of a new mobile application and podcast to reach maximum number of people in the community. Furthermore, many community members and BSB team members stated that a range of direct and indirect methods of communication using trusted sources increased community engagement. These methods included text message services, websites, newsletters and others, representing a range of different spaces. A further technological and socio-cultural influence on community engagement was the use of a range of visual forms of communication which those who did not have English as their first language could access such as the use of photographs on publicity, tiktok videos and other visual media.

## Physical influences on community engagement

Participants listed several physical influences on CEA's including the need to offer accessible venues and interpreters, and to reach a range of community members where they were through activities such as door-knocking at different times of the week to access those who worked as well as those at home during the day. Soft outcome activities were praised by participants as effective such as cook and eat sessions, healthy mum groups and walking groups around local urban spaces. A range of physical spaces were being provided for the community to gather informally in 'safe spaces' such as community buildings. Street parties and coffee mornings in targeted areas also used physical spaces well and tapped in the importance of social support and face-to-face contact with other community members so important within the socio-cultural environment. Successful CEA's adapted provision to ensure engagement during COVID-19, making use of physical spaces with smaller group activities, such as 1-2-1 walks with community members and FACE team members. The social contact these activities provided and the opportunities to leave the home environment were very important to some community members showing the overlapping importance of physical and socio-cultural environments.

## Political influences on community engagement

Political influences on community engagement fell within national, local and organisational [BSB programme] levels. Most factors discussed related to political aspects of the BSB organisation, such as governance processes. Several community members agreed that community board members had been given the opportunity to review BSB policies and take part in governance processes. It was agreed by participants that a range of community voices were represented in governance roles, examples included Roma, Refugees and Asylum seekers voices were invited to participate in the BSB partnership board. BSB team members stated that there were a range of opportunities to engage with organisations outside BSB and partners/services such as ward officers, social prescribers and parent champion groups. One community members asked for an accessible directory of support from BSB and another asked that where possible consistent, transparent information could be provided on funding criteria and other BSB process to foster greater trust from the community. This factor suggests trust, a sociocultural influence, was of central importance to CEA's. Several community members requested improved transparency within the organisation to develop complaints procedures, disseminate the organisational structure and others to show the community where they fit within the organisation. Community members also requested the use of mechanisms to allow community members to feedback their opinions about the BSB programme as it developed. Some of these were already present such as the Parent Champion groups, community champions, Community Advisory Board and Partnership board. In addition to these mechanisms the BSB team suggested that the community reference group monitoring the community engagement strategy in the future could provide an excellent feedback channel and a range of other methods such as comments boxes and feedback forms could also be used. These could be used alongside informal feedback channels after CEA's such as feedback from informal conversations during events. One community member requested that a clear communication strategy be improved by BSB to ensure that BSB explain how decisions are made to the community and where the power lies for decision-making. Furthermore, the community members requested involvement in project evaluations and BSB impact-measuring activities and that governance documents were provided in an accessible format to show key findings to community members. The FACE team stated that Project Engagement Forums offer monthly opportunities for community members to plan CEA's with project partners. Community members stated that more opportunities to engage with planning CEA's would be advantageous.

Many community members praised the opportunities they had been given by BSB to gain skills as a result of their involvement, this thereby overlapping with the individual environment. Conferences and training had provided opportunities to meet fellow volunteers and build rapport with community members, indicating influence in the socio-cultural environment. Community members and the BSB team acknowledged that more work was needed to reach hard to reach groups such as Eastern European, Roma and White British populations.

## Socio-cultural influences on community engagement

The socio-cultural environment was the most overriding influences on community engagement and many of these referred to the community specifically. We interpret the socio-cultural environment to include social factors [social support, relationships] and cultural aspects [social and cultural norms, ethnicity, religion, language]. These reflect society's values, influences and norms. Such societal and social factors greatly influenced CEA's at all levels. Our participants stated that the development of relationships between the community and community workers was key, and a priority for BSB should be increasing the capacity of the BSB workforce and sustaining strong, trusted relationships with the community to deliver health interventions and

engage the community with health services. Online provision of CEA's during the COVID-19 pandemic offered 'dip in and out' options which expanded the reach of the BSB programme to some hard to reach groups. However, central to these was trust from the community that such activities were delivered in their language or by community members/BSB workers they already knew. Recruitment of community champions had taken place from a range of different cultural groups which was seen by community members as important to have their own ethnic and cultural groups represented and 'link' workers operating between them and the BSB organisation. Warm up informal gathering in the community led by trusted community members provided valuable social interactions and were successful in bringing community members together.

Community members stated that BSB activities allowed a range of voices to be included and forums aimed at underrepresented voices, such as Roma and Dad's groups, allowed groups to self-design activities that were appropriate and culturally specific. Additionally, more general societal changes in attitudes towards community engagement had had an impact. Waves of community mindedness and increased motivation to take part in CEA's and these had optimised community volunteering and involvement during the COVID-19 pandemic. This had led to greater enthusiasm in recent months to participate in CEA's and collaborate with community champions and parent champion groups. Community members, supported by BSB team members stated the need to use a range of creative methods to reach hard to reach groups and establish initial contact with community members to develop much needed social support for them during the challenging COVID-19 pandemic. Some community members stated they liked the 'test and learn' approach taken by the BSB team and FACE team, especially during the COVID-19 pandemic, to test new engagement strategies and activities to see what was effective. FACE team workers also used 'soft intelligence' by encouraging conversations between BSB workers and the community to explore which CEA's were working well. Such trusted social interactions were key to ensuring CEA's were targeted, effective and contextually appropriate. All community members stated that the social support offered to them by the BSB team was key in helping them to feel part of the community, engaged in BSB activities and willing to come back to take part in other activities in the future. Relationships community members had developed with others in their ethnic or cultural groups [e.g. the Eastern European or Pakistani communities] which shared language or cultural norms/traditions, had increased their willingness to leave home and take part in for example antenatal classes, walking groups or English language classes. Therefore the socio-cultural environment appeared to be of overarching influence to our participants in whether they took part in CEA's.

## Discussion

Our socioecological model [Fig 1] indicates multiple layers of influence on community engagement. Our model's value is in its visual representation of the influences on community engagement behaviour in health programmes, a visualisation which is as yet under explored in the literature. In our model we use the concept of the individual at the centre followed by intermediate environment which have an influence on the behaviour of the individual and finally an all-encompassing environment at a higher level [the outer circle] which is the most overarching influential environment on all behaviours at the individual and intermediate level.

Our model, constructed of six overlapping environments; individual, economic, technological, physical, political and socio-cultural, puts the socio-cultural environment on the outer ring meaning it influences all environments within it. We categorise socio-cultural environment, drawing on elements of the socio-cultural psychological approach [47]. This approach emphasises that cultural factors such as language, social norms and social structures can play a

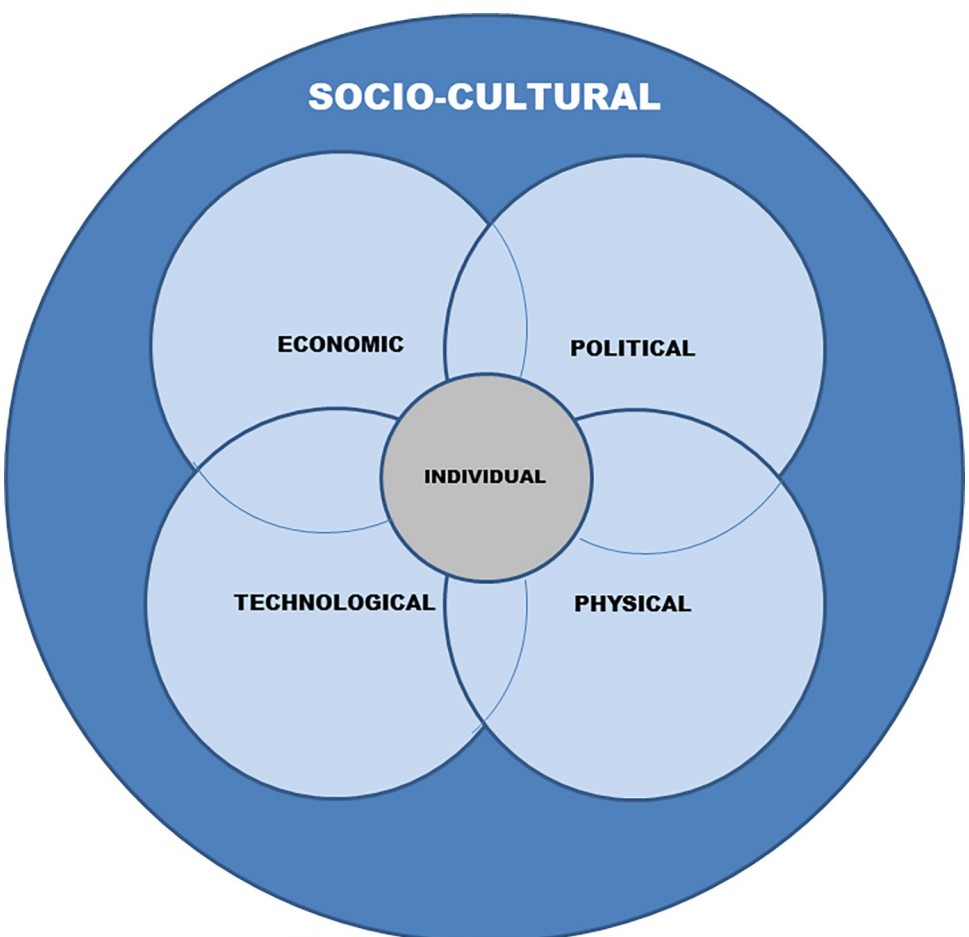

**Fig 1. Socioecological model for community engagement activities.** The individual environment representing individuals within the community is in the centre [Grey], the first level of influence. The intermediate levels of influence are economic, political, technological and physical [light blue]. The overarching level of influence is the socio-cultural environment including the community [dark blue]. The environmental influences overlap as some factors influencing community engagement in the health programme are nested within multiple environments as is the nature of SEM theory.

significant role in defining behaviours. We consider factors such as social norms [informal understandings that govern the behaviour of members of society [48]], social structures [e.g. family, communities] and cultural practices as playing an important role in behaviours. As discussed in the introduction to our setting, the location in which this study took place is diverse and multi-cultural with multiple cultural and social influences working on individual community members. The socio-cultural environment importantly incorporates 'the community' and what the community represents. Our findings have shown this to include social support, face-to-face interactions, and a wide range of cultural and social norms, ethnicities and social identities. This is illustrated by our findings showing the importance of reaching different ethnic groups such as Eastern Europeans and other social groups such as refugees and Dad's.

Whilst previous ecological models consider social or cultural environmental factors to be important in influencing other forms of behaviour such as dietary behaviour and physical activity [32, 35, 49, 50], our model explores the importance of the sociocultural environment on community engagement. Our findings state that socio-cultural influences must be considered to develop tangible, multi-faceted and contextually-appropriate ways to engage

communities in health programmes. Our approach sits in contrast to individualised approaches to community health such as monetary incentives which only consider the individual [18, 19] rather than the broader, wider, multi-faceted influences on individual behaviour.

Social support to develop skills, trust and relationships with community workers in BSB and fellow community members was shown to foster community engagement. Other research corroborates that socio-cultural determinants such as social wellbeing, trust and community identity have been important in community engagement [51]. We found that trusted sources of information were important to community members and informal consultation was the most effective when it was done by those figures such as FACE team members, or champions coming from similar ethnic or cultural backgrounds who were trusted by the community. Other research supports our findings, showing that trust is vital for establishing responsive mutual communication in community engagement [52, 53]. Our findings suggest that informal consultation, as with all community engagement activities, must include translation into different languages and be culturally appropriate, supporting other research which has found socio-cultural context to be a key consideration in developing community interventions [35, 54]. Such informal consultation and multi-lingual and multi-cultural activities by community organisations like BSB can lead to improved community readiness in specific minority groups, as has been found in Roma communities in Bradford [55] and other minority groups around the world [56].

The physical, political, economic and technological environments all influenced community engagement behaviour. The physical environment defined aspects such as physical spaces where community engagement took place, these included trusted community buildings as well as opportunities for physical face-to-face interactions such as door-knocking. Other research has found that communal spaces have positive effects on community engagement and can foster social capital and place attachment [57]. Within the political environment BSB organisational policies and governance structures, and how accessible they were to the community, were important in influencing behaviour. We found that political processes within the BSB organisation needed to be made more transparent and feedback channels made clearer and stronger for all in the community. These political influences were highly influenced by socio-cultural environmental factors including the need to be inclusive and allow hard to reach voices to be heard. Community agency in governance processes has been found to be key in improving engagement and community participation [58]. Furthermore, community engagement should reflect national democratic processes, with programmes such as BSB aiming to reflect democratic values in allowing community voices to drive and guide community engagement activities [59].

The economic environment, influenced by the sociocultural environment, and sometimes overlapping with political or technological environments, impacted upon some community engagement behaviour. These included community members' request for BSB to ensure funding for CEA's was allocated to local organisations in a sustainable way, and allow funding criteria to be made readily available for the community to scrutinise. Other factors, such as the economic status of individuals influenced the amount of time they had to spend on CEA's and their ability to buy technological equipment that they might need to take part in virtual CEA's during the COVID-19 pandemic. We found that community members without access to skills to use technology or the technology itself should not be excluded from CEA's. These community members must also be catered for as digital inequalities are particularly apparent in ethnic minority groups and those from low socio-economic backgrounds [60–63]. We found that a range of methods were preferred in both physical and virtual/technological environments to generate maximum community engagement with visual forms of communication being important for those minority communities who may not have English as their first language.

Our findings suggest that community engagement should consist of a dual offering adopting a blended approach including virtual and face-to-face offerings to cater for all community needs. Furthermore, our findings suggest that social support should be used within the community to develop programmes to improve technological skills and language abilities to allow for more effective community engagement [64, 65].

The individual level of influence affected community engagement in some aspects, such as the individual's language or technological skills, or time to spend outside the home on CEA's. However, we found that all individual factors were greatly influenced by surrounding environments, most prevalently the socio-cultural environment. This suggests that individual actions should not be seen in terms of individual behaviour change and that individual capabilities and motivations are broadly influenced by multiple environments in society [20, 27, 30–32, 35, 56] with socio-cultural influences playing an overarching role [35, 54]. Our findings further support socioecological systems theory acknowledging the central influence of beliefs and ideologies across society whilst considering the interconnections and dependencies amongst community members [66]. A whole communities approach allows us to consider the influences on the community across multiple levels and environments [66–68]. Our findings add to a growing body of literature which argue that community-led solutions can address the social and structural determinants of health [9].

The context of the COVID-19 pandemic has led to an increase in community mindedness and increased motivation demonstrated in Bradford to optimise community volunteering [69]. In the context of the COVID-19 pandemic, community engagement has become increasingly important as vaccine hesitancy becomes an issue amongst some communities needing tackling and strategies to engage hard to reach communities to address health inequalities is vital [70, 71].

The strength of SEM's is that they can be used to understand a range of factors which influence people's behaviour. SEMs such as ours can be used to develop strategies which can span multiple environments. An example of this is identifying community members who lack technological skills and equipment to engage in certain community engagement activities, and then developing training, providing equipment and ensuring social support is given to develop individual's technical skills and support them in their engagement in health programmes such as BSB. Such strategies cross technological, economic, socio-cultural and individual environments in our model to represent a complex intervention which considers multiple influences on behaviour.

In our study, we have used a case study, a valuable form of research [72], to develop a SEM which allows us to develop a more nuanced understanding of the reasons why communities may or may not engage with health programmes such as BSB. Using this knowledge we can begin to develop interventions and strategies which consider all the influences on the individual illustrated in our model to tackle lack of engagement in health programmes. Such strategies go beyond the individualised methods of incentivising individual behaviour and see community engagement in health programmes as multi-faceted and multi-factorial.

Our study is not without limitations. Our service design process only represented community engagement in three wards in one city in the UK. Our workshops included a limited number of community members, and some groups in the community were under or unrepresented. Due to COVID-19 restrictions our workshops were held virtually and could have excluded some members of the community who did not have access to the internet or necessary technologies to take part. We also acknowledge that there are limitations with relying on the data collected from the service design process. Our synthesis of the key themes which formed the environments in the socio-ecological model was based on data which was not explicitly designed to inform such a model. The questions asked and discussions had

during the workshops were focused on the formation of a logic model for community engagement with BSB and not specifically on the formation of our socio-ecological model. Ideally we would conduct more workshops specifically aimed at further developing, fleshing out and strengthening the socio-ecological model we have proposed. We cannot generalise our findings for other geographical locations where communities may involve different community engagement behaviour with different environmental influences Nonetheless, we consider that putting together these insights into a socio-ecological model allowed us, and hopefully will allow others, to understand barriers and enablers to community engagement in a more systematic way. Further testing would be required to investigate whether the same principles apply to other geographical, ethnic and socio-economic groups.

## Conclusion

Our study represents the first time that community engagement behaviour has been considered in the form of a socioecological model, exposing the potential of taking a socioecological lens on community engagement activities to develop multi-faceted solutions to tackle health inequalities. Our socioecological model represents an opportunity to visualise the influences on community engagement in an underserved urban area so that community engagement activities can be more effectively developed which consider multiple environmental influences. Central to future community engagement activities should be the acknowledgment and incorporation of the socio-cultural environment. Developing relationships with the community built on trust and around provision of social support are vital. Consideration of the impact of economic, technological, political, physical and individual influences are also important to ensure community engagement activities are multi-dimensional and consider social and structural determinants of health [9]. We therefore hope our insights will allow others to understand the barriers and enablers to community engagement in health in a more systematic way. We advocate testing our socioecological model with a range of different communities within the UK and globally, to explore to what extent our model is applicable in a range of different community contexts.

## Supporting information

**S1 Table. Workshop guide for support standard.**
(DOCX)

**S2 Table. Full list of influences on community engagement discussed by workshop participants and their socioecological environments.**
(DOCX)

## Acknowledgments

This project was only possible because of the enthusiasm and commitment of the members of the Community Reference Group. We are grateful to all the participants, the Community Reference Group, the Better Start Bradford partnership and staff, BSB projects, health professionals and researchers who have helped to make this project possible.

## Author Contributions

**Conceptualization:** Lizzie Caperon.

**Data curation:** Lizzie Caperon, Fiona Saville.

**Formal analysis:** Lizzie Caperon.

**Investigation:** Lizzie Caperon.

**Methodology:** Lizzie Caperon.

**Project administration:** Fiona Saville, Sara Ahern.

**Resources:** Fiona Saville.

**Writing – original draft:** Lizzie Caperon.

**Writing – review & editing:** Lizzie Caperon, Fiona Saville, Sara Ahern.

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
