## [Decision Letter · Decision Letter 0]

7 Mar 2022

PONE-D-21-26119Developing a socio-ecological model for community engagement in an ethnically diverse and deprived urban area; a coproduction evaluationPLOS ONE

Dear Dr. Caperon,

Thank you for submitting your manuscript to PLOS ONE. After careful consideration, we feel that it has merit but does not fully meet PLOS ONE’s publication criteria as it currently stands. Therefore, we invite you to submit a revised version of the manuscript that addresses the points raised during the review process. Two reviewers and I have reviewed your manuscript, and have identified several issues that should be addressed before it could be further considered for publication. I have pulled what I view are the primary reviewer concerns and my own and enumerated them here.It is difficult to evaluate how your study fits into existing knowledge. A more rigorous, thorough review of existing literature on community engagement behaviors, particularly from sociological literature and that of relevant applied fields, would be helpful.Some methodological details and constraints should be addressed. Namely, how were candidate participants identified in the first place? And what are the limitations associated with your reliance on workshops? Was the target population for the BSB programme omitted from your workshops for the very reasons they are constrained from CEA behaviors? Were the same 10 community members at all nine of the workshops, or a different 10 each time? If the same 10, how does this sample size or repeated engagement of the same people affect your findings, either for better or worse?A more detailed explanation is warranted of the ethnic, economic, and cultural backgrounds of the study areas and participant communities. Correspondingly, terminology such as “ethnically diverse” and “deprived”, if retained, should be defended as the accepted and appropriate terminology for the study community.It does not seem that this study really involves co-production. What was co-produced: the BSB programme itself, or your research about it? Or, perhaps neither is co-produced, and you might instead consider different terminology? The research methods reported in this manuscript do not indicate co-production.The BSB programme isn’t much described. Since this paper evaluates participation in the BSB programme, more information about the programme is warranted. While the programme seems to target children’s health, the manuscript doesn’t take up that theme after initially introducing it in the “setting” section.The manuscript claims to develop a model, but the emergence of the model from the research is unclear. The model appears to consist largely of circles containing the “theme” or “environment” labels. It is not annotated nor well-described. It is not evident how those themes or environments were arrived at. Were they established a priori, or did they emerge as an outcome of considering the workshop transcripts? Moreover, is the size, position, and nesting or overlap of the circles significant? What does their spatial organization in the diagram convey? And, what is the role or significance of the floating words (“language”, “Socio cultural norms”, etc.)? Please clarify what makes this diagram a socioecological “model” of community engagement behaviour, and how one would use the model.On a related note, table 1 is unuseful to the reader. It is not easily readable, and is largely redundant with material in the main text. Perhaps a more synthetic table with key take-home points could be made, and the table contents as they exist now be moved to supplemental information.The manuscript is riddled with small errors of grammar and punctuation.Please indicate the mechanism by which your data will be made available.A heavily revised manuscript could be considered for possible publication if you are able to address these concerns.

We look forward to receiving your revised manuscript.

Kind regards,

David B. Lewis, Ph.D.

Academic Editor

PLOS ONE

“This service design process received funding from the Big Lottery Fund as part of the A Better Start programme. The Big Lottery Fund have not had any involvement in the design or writing of the paper.”

“Bradford Institute for Health Research received funding for this service design process the Big Lottery Fund, UK (https://www.tnlcommunityfund.org.uk/ ) as part of the A Better Start programme.

Reviewers' comments:

Reviewer's Responses to Questions

**Comments to the Author**

1. Is the manuscript technically sound, and do the data support the conclusions?

Reviewer #1: Partly

Reviewer #2: No

2. Has the statistical analysis been performed appropriately and rigorously? 

Reviewer #1: N/A

Reviewer #2: N/A

3. Have the authors made all data underlying the findings in their manuscript fully available?

Reviewer #1: No

Reviewer #2: No

4. Is the manuscript presented in an intelligible fashion and written in standard English?

Reviewer #1: Yes

Reviewer #2: Yes

5. Review Comments to the Author

Reviewer #1: Remarks to the Author:

*It is important to note that I am an expert in network science for complex social systems and interest in social mobility and gathering. However, I’m not an expert on sociology.

A. Originality and the key results

As far as I know, it seems to be the first application of socio-ecological analysis to community engagement in an ethnically diverse and deprived urban area. The author introduces the socio-ecological model with multiple layers of influence on community engagement activities (CEA) in six environmental factors: individual, political, physical, technological, economic, and socio-cultural. Each factor is comprehensively analyzed and presented with its associated influence as well as its own influence on community engagement.

The author highlights the importance of the overall influences of the socio-cultural environment which include social support, trust, relationships in communities, and ethnicity or language. At the same time, the author presents multi-factored aspects needed to ensure community engagement activities such as time spent on CEA, transparency through democratic process, and access to technology or communal spaces.

Finally, the author suggests that considering multiple environmental influences and incorporation of the socio-cultural environment can be more effectively developed the community engagement activities.

B. Data and model coverage

The co-production process that has been analyzed in this manuscript only represents the communities within three electoral wards that are considered deprived and have diverse ethnic populations. Also, workshops have some limitations in the number of community members or the existence of under or unrepresented groups as mentioned in the manuscript.

Due to the specificity of these data, further detailed explanations of the ethnic, economic, and cultural backgrounds of the restricted areas or participated communities are needed to avoid the error of hasty generalization and to specify the scope of application of the model.

C. Suggested improvements

The framework analysis and developed matrix structure of key socio-ecological themes are adopted to analyze the data. Although framework analysis lists environments that influence community engagement, the evidence in the results is insufficiently descriptive so statistical analysis or regression methods are recommended to support the results.

Reviewer #2: This paper provides a general sociological look at the complexity of community engagement related to a health program, offering insights into the interconnected and multifaceted aspects of community life that might limit or enable engagement among diverse individuals. The use of community focus groups and workshops seems appropriate to the research questions. I have some concerns with the framing, novelty, contribution, and overall clarity of the article.

Overall, I found the framing of the methods to be confusing. Either the authors are doing a sociological evaluation of a “coproduced” program, or they are trying to co-produce research findings within a sociological model (but not really using any coproduction methods), but the objectives are unclear and mismatched with the methods and findings.

The introduction needs to provide a broader view of the need for this sort of work in terms of theoretical or methodological developments, criticism, evaluation, or novel contributions.

You are clearly applying a co-produced research method for good reason, but why this would be useful or informative to the journal’s audience is unconvincing. There is really limited discussion or nuance around the theory of co-production here, or health related co-production work. Further, the language you use to describe the community is so vague and at patronizing. Did the participants in this coproduced project think of themselves as “ethnically diverse and deprived”? Which ethnicities? Do sociologists use the term deprived? If so, do you have a citation? Or some quantitative data to inform us of the demographics and trends in this area relative to your terms?

It is unclear what sort of entity is running the project, relative to the community leadership and the various layers of governance and organizing that could be going on in the area. Is this a university? A local or national government? What role did the community have in security the funding, or setting project goals?

I would suggest finding an alternative to the term “stakeholder.”

I would suggest using direct language and taking up an active voice to more clearly indicate various actions and sources of agency and activity in your narrative.

How interesting that program design is not considered research and does not require ethics clearance. I understand this is a strange “no-researcher-land” where co-produced research is concerned, however I’m also concerned that you have not cited the large body of ethics research, particularly ethics for working with underserved communities of color in research, in your ethics section. You have not outlined efforts to provide information or reciprocity back to your community participants in any form. If the Scottish standards for workshops of this type provide some details here, those should be explicitly documented in the methods.

Your participant recruitment methods are very unlearn.

After reading your analysis methods, it is quite clear that this is not coproduction, but focus group research conducted by the research team using thematic analysis.

Did you perhaps also refer to the literature while conducting this analysis? If so, which theoretical body of work informed your assessment of the data? Do you have any citations to suggest evidence of how you enhanced the trustworthiness and validity, or relationship to theoretical grounding, in your work?

6. PLOS authors have the option to publish the peer review history of their article (what does this mean?). If published, this will include your full peer review and any attached files.

Reviewer #1: No

Reviewer #2: No

---

## [Author Response · Author response to Decision Letter 0]

20 Apr 2022

Our detailed response to reviewers is presented in a table in the attached document named 'Response to Reviewers'. Please find the text of that response below (without the table formatting). Many thanks.

Dear Plos One team, 

Please find our response to reviewer and editor comments below.

Compiled editor comments Author response to compiled editor comments

1. It is difficult to evaluate how your study fits into existing knowledge. A more rigorous, thorough review of existing literature on community engagement behaviors, particularly from sociological literature and that of relevant applied fields, would be helpful. Thank you for this observation. On re-reading the introduction we acknowledge that our aims and grounding in the literature were not clear. We have undertaken a thorough re-writing of the introduction to better set our study within the literature, making it clear that we aim to explore how a socio-ecological model can explore more multi-faceted solutions to community engagement in health programmes. 

We would like to thank reviewer 1 for their comments regarding the originality of the study, indicating that it is well placed to explore an underexplored area:

‘A. Originality and the key results

As far as I know, it seems to be the first application of socio-ecological analysis to community engagement in an ethnically diverse and deprived urban area. The author introduces the socio-ecological model with multiple layers of influence on community engagement activities (CEA) in six environmental factors: individual, political, physical, technological, economic, and socio-cultural. Each factor is comprehensively analyzed and presented with its associated influence as well as its own influence on community engagement.

The author highlights the importance of the overall influences of the socio-cultural environment which include social support, trust, relationships in communities, and ethnicity or language. At the same time, the author presents multi-factored aspects needed to ensure community engagement activities such as time spent on CEA, transparency through democratic process, and access to technology or communal spaces.

Finally, the author suggests that considering multiple environmental influences and incorporation of the socio-cultural environment can be more effectively developed the community engagement activities.’

As we now explain in the introduction we argue that socio-ecological modelling is the best conceptual tool to gather the common themes we were identifying from a service design process in our study area with underserved communities in Bradford, UK. For example, our introduction states:

‘By developing this socio-ecological model we hope that community engagement in health programmes can be understood more systematically and holistically considering wider social determinants, and therefore the barriers to community engagement in health programmes can be addressed in multifaceted ways which take into account the social determinants of health. Such an understanding holds the potential to look beyond individualised conceptualisations of behaviour but rather takes into account a multitude of social and cultural influences.’

2. Some methodological details and constraints should be addressed. Namely, how were candidate participants identified in the first place? 

And what are the limitations associated with your reliance on workshops? 

Was the target population for the BSB programme omitted from your workshops for the very reasons they are constrained from CEA behaviors? 

Were the same 10 community members at all nine of the workshops, or a different 10 each time? If the same 10, how does this sample size or repeated engagement of the same people affect your findings, either for better or worse? Thank you for your observations – as a result of them we have realised that our methods were not clear and we apologise for this. We have described our methods much more clearly now. For example, in the introduction under ‘Setting’ we state that as we were co-producing the logic model for CE (community engagement) during workshops with community members as part of a separate, initial piece of research, we started to notice common themes on the influences of behaviour for community engagement. It was out of these observations that we decided to put these themes together into a socio-ecological model to more systematically and holistically explain why people do or do not engage in community health programmes:

‘ In October 2020 we set out to develop a logic model for community engagement in the BSB programme as part of the programme service design process. This co-production process involved running nine workshops with community members including pregnant women and parents of children under 4 which the BSB targets, FACE team members, health professionals, researchers and BSB staff. These workshops explored the barriers to the community engaging in the programme as well as enablers which increased community engagement in the programme. As we were co-producing the logic model for community engagement we started the notice common themes on the influences of behaviour for community engagement coming out of the workshops. As a result we decided to put these themes together into a socio-ecological model to more systematically and holistically explain why people do or do not engage in community health programmes. Therefore, this theoretical paper documents the results of a service evaluation which took place as part to review the existing service design of Community Engagement across the BSB programme at the midpoint of programme delivery.’

Therefore, participants were identified as part of the initial study. We have explained in the methods under ‘Participants’ how participants were identified for the initial study:

‘Our participants had all been recruited to take part in the service design co-production process to evaluate the BSB Programme. These participants had been identified through discussion with members of the BSB team, specifically the Family and Community Engagement [FACE] team...’

We have clarified in the ‘participants’ section that:

‘We ensured that we recruited pregnant women and parents with children under the age of 4 who are the target population for the BSB programme.’

We have edited the ‘recruitment and consent’ section to explain that:

‘Three community members attended all nine workshops, however many attended two or three of the workshops, and the composition of the group in the workshops changed every week. The total number of participants over all workshops was 25.’

We have explained in our limitations section (penultimate paragraph of the discussion) that:

‘Our service design process only represented community engagement in three wards in one city in the UK. Our workshops included a limited number of community members, and some groups in the community were under or unrepresented. Due to COVID-19 restrictions our workshops were held virtually and could have excluded some members of the community who did not have access to the internet or necessary technologies to take part. We cannot generalise our findings for other geographical locations where communities may involve different community engagement behaviour with different environmental influences. Further testing would be required to investigate whether the same principles apply to other geographical, ethnic and socio-economic groups.’

We have added to this section that we acknowledge the limitations of relying on the workshops for the formation of our socio-ecological model:

‘We also acknowledge that there are limitations with relying on the data collected from the service design process. Our synthesis of the key themes which formed the environments in the socio-ecological model was based on data which was not explicitly designed to inform such a model. The questions asked and discussions had during the workshops were focused on the formation of a logic model for community engagement with BSB and not specifically on the formation of our socio-ecological model. Ideally we would conduct more workshops specifically aimed at further developing, fleshing out and strengthening the socio-ecological model we have proposed. Nonetheless, we consider that putting together these insights into a socio-ecological model allowed us, and hopefully will allow others, to understand barriers and enablers to community engagement in a more systematic way.’

3. A more detailed explanation is warranted of the ethnic, economic, and cultural backgrounds of the study areas and participant communities. Correspondingly, terminology such as “ethnically diverse” and “deprived”, if retained, should be defended as the accepted and appropriate terminology for the study community. Thank you, we agree that a more detailed explanation of the backgrounds of the study areas was warranted. We have included a description of each of the three wards BSB serves in the Setting section:

‘Our study takes place with a community in an economically underserved urban area within three electoral wards in which the Better Start Bradford programme operates in Bradford, in northern England. Bradford was hit particularly hard by the COVID-19 pandemic (10). The wards BSB serves are considered amongst the most underserved in England and have populations made up of many different ethnicities (39). Bowling and Barkerend is the one ward which BSB serves. It has a total population of 22,200, 11.2% of houses in the ward are overcrowded and 29.2% of the population are under 16. Life expectancy in the ward is lower than the district average (73.9 for men and 78.6 for women) and the ward is ranked 3rd out of 30 in the District for the 2019 Index of multiple deprivation where 1 is the most deprived. 42.7% of the population in the ward are white and 32.9% are Pakistani, with the remaining 24.3% made up of a range of other ethnicities. The dominant religion in the ward is Muslim (45.8%) with Christian (29.7) and no religion (14.6%) second and third respectively (40). Bradford Moor is the second ward BSB serves with a population of 21,310 it is similar in size to Bowling and Barkerend but more houses (17.3%) are overcrowded in the ward. Like Bowling and Barkerend, life expectancy in the ward is lower than the district average (74 for men and 80 for women). Bradford Moor is ranked 4 out of 30 wards in the District for the index of multiple deprivation. Demographically, 63.9% of the population of the ward are Pakistani and 17.3% are White with the remainder a mixture of different ethnicities. 72.8% of the ward are Muslim, 13% Christian and the remaining 14.2% belong to religions or have no religion(41). Little Horton is the final district BSB serves with a population of 23,340 it is the largest of the three wards. 14.1% of homes are overcrowded and like the other two wards; life expectancy is below the district average at 76.3% for men and 82.2 for men. Little Horton is ranked 2nd of the 30 Wards in the District for the 2019 index of multiple deprivation. The majority of the population is Pakistani (48.5%) with 28.8% white. The main religion in the ward is Muslim (58%) with 24.7% Christian and the remaining 17.3% a mixture of other religions or with no religion (42).’ 

Thank you for your valuable observations regarding language, as researchers we must always be open to revising the language we use to describe populations. We have replaced all references to deprived with underserved and we have removed reference to ethnically diverse to describe our study population.

4. It does not seem that this study really involves co-production. What was co-produced: the BSB programme itself, or your research about it? Or, perhaps neither is co-produced, and you might instead consider different terminology? The research methods reported in this manuscript do not indicate co-production. From your comments it’s clear that our description of co-production was confusing, we apologise for this. We meant to explain that the logic model process from which our data was derived was co-produced. We have removed reference to co-production throughout the article to make it clearer except in relation to the logic model process.

We have explained in the setting section of the introduction that the co-production was only in relation to the logic model process and removed co-production from the title of this article. 

5. The BSB programme isn’t much described. Since this paper evaluates participation in the BSB programme, more information about the programme is warranted. While the programme seems to target children’s health, the manuscript doesn’t take up that theme after initially introducing it in the “setting” section. Thank you we have provided more detail on the description of the BSB programme. We have now clarified that the paper doesn’t evaluate participation in the BSB programme. Rather it explores the barriers and enablers to community engagement, using the BSB programme as a case study.

‘The Better Start Bradford [BSB] Programme is funded by the National Lottery Community Fund over 10 years 2015-2025. The programme aims to improve the health outcomes of children by commissioning projects and services aimed at pregnant women and families with children under the age of 4. Engaging families with these projects and services, as well as with key programme messages promoting healthy child development, is therefore crucial to the success of the programme. The BSB programme includes community members in a range of governance roles and in engagement roles. The Family and Community Engagement (FACE) team are link workers employed by BSB to develop community partnerships, support parent led group activities and recruit volunteers to the BSB Programme Community champions are members of the community who promote the BSB programme activities.’

And

‘This paper does not provide an evaluation of participation in the BSB programme, rather it explores the barriers and enablers to community engagement in the BSB programme through a socio-ecological lens, using the BSB programme as a case study.’

We have provided detail on the role of the BSB programme in Bradford and have also added a more detailed explanation of what the logic model process (from which our data was derived) was aiming to achieve in the Setting section:

‘In October 2020 we set out to develop a logic model for community engagement in the BSB programme as part of the programme service design process. This co-production process involved running nine workshops with community members including pregnant women and parents of children under 4 which the BSB targets, FACE team members, health professionals, researchers and BSB staff. These workshops explored the barriers to the community engaging in the programme as well as enablers which increased community engagement in the programme. As we were co-producing the logic model for community engagement we started the notice common themes on the influences of behaviour for community engagement coming out of the workshops. As a result we decided to put these themes together into a socio-ecological model to more systematically and holistically explain why people do or do not engage in community health programmes.’

6. The manuscript claims to develop a model, but the emergence of the model from the research is unclear. The model appears to consist largely of circles containing the “theme” or “environment” labels. It is not annotated nor well-described. It is not evident how those themes or environments were arrived at. Were they established a priori, or did they emerge as an outcome of considering the workshop transcripts? Moreover, is the size, position, and nesting or overlap of the circles significant? What does their spatial organization in the diagram convey? And, what is the role or significance of the floating words (“language”, “Socio cultural norms”, etc.)? Please clarify what makes this diagram a socioecological “model” of community engagement behaviour, and how one would use the model. Thank you for this comment, we apologise for the lack of clarity with describing how the model emerged. We have now provided a much more detailed discussion of Socio-ecological models in the introduction to set context. We have also provided a detailed explanation of how the environments in the model were developed partially a priori and partially a posteriori in the results section. This explanation links to the socio-ecological literature:

‘Our data analysis began with a priori codes from the socio-ecological model [SEM] literature. These codes were formed from the initial theory behind Brofenbrenner’s first socio-ecological model which were nesting circles that placed the individual in the centre surrounded by various influencial systems (19). The microsystem closest to the individual contains the strongest influences and incorporates the interactions and relationships of the immediate surroundings. The outer rings of the models or outer environments have traditionally represented environments which have interactive forces on the individual such as community contexts or social networks(46). SEM models illustrate that health behaviours are affected by the interaction between the characteristics of the individual, community and physical, social and political components. 

Therefore our a priori codes drew on existing models, particularly models which place socio-cultural influences and community as an overarching macro influence on behaviours (33). We began loosely therefore with three levels of codes – individual, intermediate and higher. However, as we began to code the data we discovered that specific environments were present within the intermediate and higher levels which were specific to the community engagement behaviour. Following this we began to formulate the dominant ‘influences’ or environments such as political, economic, physical and technological influences’

In this section we have now also provided a table which shows the levels of influence and themes which emerged from the analysis (Table 1). For clarity we have edited the SEM to remove any floating words and ensure that the levels of influence in the model are clearly labelled and described. 

‘Analysis of our data therefore, both a priori and a posteriori, led to thematic representation under six environments; individual, political, physical, technological, economic and socio-cultural environments. Within each environment we coded specific aspects such as ‘time to take part in activities’ (individual) or ‘allocation of funding’ (economic). We found that many codes overlapped several environments, as socio-ecological modelling encourages, to show multi-faceted influences on behaviour. The full list of codes and the environments these fit within can be found in supporting information. Our list of levels of influence and example codes can be found in Table 1.’

We have now provided detailed explanation of size, position and nesting/overlap in the caption below the figure of the model. 

We have also strengthened our discussion of the model in the discussion section;

‘Our findings state the need for socio-cultural influences to be considered develop tangible, multi-faceted and contextually-appropriate ways to engage communities in health programmes. Our approach sits in contrast to individualised approaches to community health such as monetary incentives which only consider the individual(15,16) rather than the broader, wider, multi-faceted influences on individual behaviour.’

We further elaborate on how our SEM model can be used in our discussion section:

‘The strength of SEM’s is that they can be used to understand a range of factors which influence people’s behaviour. SEMs can be used to develop strategies which can span multiple environments such as identifying community members who lack technological skills and equipment to engage in certain community engagement activities, and then developing training, providing equipment and ensuring social support is given to develop individual’s technical skills and support them in their engagement in health programmes such as BSB. Such strategies cross technological, economic, socio-cultural and individual environments in the model to represent a complex intervention which considers multiple influences on behaviour. 

In our study, we have developed a SEM which allows us to develop a more nuanced understanding of the reasons why communities may or may not engage with health programmes such as BSB. Using this knowledge we can begin to develop interventions and strategies which consider all the influences on the individual illustrated in our model to tackle lack of engagement in health programmes. Such strategies go beyond the individualised methods of incentivising individual behaviour and see community engagement in health programmes as multi-faceted and multi-factorial.’

7. On a related note, table 1 is unuseful to the reader. It is not easily readable, and is largely redundant with material in the main text. Perhaps a more synthetic table with key take-home points could be made, and the table contents as they exist now be moved to supplemental information. Thank you, we agree that Table 1 was long and detailed. We have moved this table to supplemental information. We have now created a new table 1 which illustrates in a much more succinct and clear manner the levels of influence developed in our analysis with examples from the original table. 

8. The manuscript is riddled with small errors of grammar and punctuation. Thank you for this observation, we apologise for the errors and we have re-read the manuscript carefully to correct them. 

9. Please indicate the mechanism by which your data will be made available. Thank you for this observation. Our study is a qualitative study. PLOS One states on in your instructions for qualitative data the following:

‘For studies analyzing data collected as part of qualitative research, authors should make excerpts of the transcripts relevant to the study available in an appropriate data repository, within the paper, or upon request if they cannot be shared publicly. If even sharing excerpts would violate the agreement to which the participants consented, authors should explain this restriction and what data they are able to share in their Data Availability Statement.’

The process from which data in this article is derived was an exercise in service design which would go on to inform future service evaluation of community engagement in the Better Start Bradford programme. As such it did not require formal ethical review approval (HRA decision 60/88/81). All participants in the workshops provided written consent for their data to be used anonymously and for findings derived from the service design process to be published in this article. For this reason, no identifying personal information was obtained or recorded for the purposes of research or any other use. The data which the research team holds is in the form of informal notes from the workshops which took place. This data is not in the form of formal transcripts but these researcher notes can be made available upon request. Detailed data gathered from the study showing the key themes discussed can be found in supporting information 2. This forms our minimal data set. Our Data Availability Statement will be updated accordingly. 

Reviewer 1 comments Response to Reviewer 1 comments

*It is important to note that I am an expert in network science for complex social systems and interest in social mobility and gathering. However, I’m not an expert on sociology.

A. Originality and the key results

As far as I know, it seems to be the first application of socio-ecological analysis to community engagement in an ethnically diverse and deprived urban area. The author introduces the socio-ecological model with multiple layers of influence on community engagement activities (CEA) in six environmental factors: individual, political, physical, technological, economic, and socio-cultural. Each factor is comprehensively analyzed and presented with its associated influence as well as its own influence on community engagement.

The author highlights the importance of the overall influences of the socio-cultural environment which include social support, trust, relationships in communities, and ethnicity or language. At the same time, the author presents multi-factored aspects needed to ensure community engagement activities such as time spent on CEA, transparency through democratic process, and access to technology or communal spaces.

Finally, the author suggests that considering multiple environmental influences and incorporation of the socio-cultural environment can be more effectively developed the community engagement activities. Thank you for these positive comments and observations, we really appreciate them. 

B. Data and model coverage

The co-production process that has been analyzed in this manuscript only represents the communities within three electoral wards that are considered deprived and have diverse ethnic populations. Also, workshops have some limitations in the number of community members or the existence of under or unrepresented groups as mentioned in the manuscript.

Due to the specificity of these data, further detailed explanations of the ethnic, economic, and cultural backgrounds of the restricted areas or participated communities are needed to avoid the error of hasty generalization and to specify the scope of application of the model. Thank you for these observations. As explained above we have expanded our limitations section and substantially strengthened our section on the detail of the three wards which our study covers in Bradford. (see Setting section)

In our limitations section we state that:

‘We also acknowledge that there are limitations with relying on the data collected from the service design process. Our synthesis of the key themes which formed the environments in the socio-ecological model was based on data which was not explicitly designed to inform such a model. The questions asked and discussions had during the workshops were focused on the formation of a logic model for community engagement with BSB and not specifically on the formation of our socio-ecological model. Ideally we would conduct more workshops specifically aimed at further developing, fleshing out and strengthening the socio-ecological model we have proposed.’

C. Suggested improvements

The framework analysis and developed matrix structure of key socio-ecological themes are adopted to analyze the data. Although framework analysis lists environments that influence community engagement, the evidence in the results is insufficiently descriptive so statistical analysis or regression methods are recommended to support the results. We appreciate your suggestions. However as this is a qualitative study, we don’t believe our sample size would be large enough for statistical analysis. Furthermore the aim of the paper is to go in depth not search for representativeness or generalisability as statistical analysis or regression methods would do. As this well cited paper states (Flyvbjerg, 2006), case study research such as ours has merit and can strengthen our understanding of our social environment: https://journals.sagepub.com/doi/abs/10.1177/1077800405284363

Reviewer 2 comments Author response to reviewer 2 comments

This paper provides a general sociological look at the complexity of community engagement related to a health program, offering insights into the interconnected and multifaceted aspects of community life that might limit or enable engagement among diverse individuals. The use of community focus groups and workshops seems appropriate to the research questions. I have some concerns with the framing, novelty, contribution, and overall clarity of the article. Thank you, we believe we have fully addressed these issues now. We have changed the description of our methodology to that of workshops. We have also reframed our research question, explained in depth how our approach is novel and that it contributes to an as yet unexplored area of research. We have thoroughly rewritten large sections of the article and ensured a clear message runs throughout it. 

Overall, I found the framing of the methods to be confusing. Either the authors are doing a sociological evaluation of a “coproduced” program, or they are trying to co-produce research findings within a sociological model (but not really using any coproduction methods), but the objectives are unclear and mismatched with the methods and findings. Please see our response to Editorial comment 4. Apologies for the confusion, made have made our methods clearer and coproduction has been largely removed from the article.

The introduction needs to provide a broader view of the need for this sort of work in terms of theoretical or methodological developments, criticism, evaluation, or novel contributions. Please see response to editorial comment 1. We have now thoroughly rewritten the introduction. The introduction now places this research in the context of the broader research question – increasing community involvement in health programmes. We explain how this paper tries to move beyond considering individual incentives to look more broadly at wider environments and factors which can influence participation in community health programmes. 

You are clearly applying a co-produced research method for good reason, but why this would be useful or informative to the journal’s audience is unconvincing. There is really limited discussion or nuance around the theory of co-production here, or health related co-production work. Further, the language you use to describe the community is so vague and at patronizing. Did the participants in this coproduced project think of themselves as “ethnically diverse and deprived”? Which ethnicities? Do sociologists use the term deprived? If so, do you have a citation? Or some quantitative data to inform us of the demographics and trends in this area relative to your terms? Please see response to editorial comment 4 regarding our use of co-production. We have removed the terms ethnically diverse and deprived. We have provided detailed statistics about the ethnicities in the research area. See also our response to editorial comment 3 above.

It is unclear what sort of entity is running the project, relative to the community leadership and the various layers of governance and organizing that could be going on in the area. Is this a university? A local or national government? What role did the community have in security the funding, or setting project goals? We have clarified that the SEM (socio-ecological model) wasn’t coproduced only the logic model was coproduced. See editorial comment 4 above.

We have clarified that the research is situated within the BSB (Better Start Bradford) programme and the researchers running the project are from the BSBIH (Better Start Bradford Innovation Hub) which is the evaluation arm of the BSB programme located in BRI (Bradford Royal Infirmary) and BIHR (Bradford Institute for Health Research). 

We have also clarified where funding for BSB came from in the Funding Declaration.

I would suggest finding an alternative to the term “stakeholder.” More specific terms have been used directly after the term stakeholder is used (once in the article) to qualify what we mean by the word.

I would suggest using direct language and taking up an active voice to more clearly indicate various actions and sources of agency and activity in your narrative. Thank you, we have gone through the article and ensured that we are using a more active voice to denote agency. 

How interesting that program design is not considered research and does not require ethics clearance. I understand this is a strange “no-researcher-land” where co-produced research is concerned, however I’m also concerned that you have not cited the large body of ethics research, particularly ethics for working with underserved communities of color in research, in your ethics section. You have not outlined efforts to provide information or reciprocity back to your community participants in any form. If the Scottish standards for workshops of this type provide some details here, those should be explicitly documented in the methods. As detailed in response to editorial comment 4 above, we have clarified that we are not using co-production to create our SEM. Rather we have taken data from what was a co-production process to create a logic model as part of a service design process. We have taken advice from HRA and they confirm that the data generated from the service design process did not require formal ethical review approval (HRA decision 60/88/81). Therefore the ethics relating to the data generated from the logic model process were sound. We did not use any identifying data in our process and all participants in the original service design proves provided written consent for their data to be used and findings published.

Your participant recruitment methods are very unlearn. Thank you for this observation, we have clarified our recruitment methods, please see our response to editorial comment 2 above. 

After reading your analysis methods, it is quite clear that this is not coproduction, but focus group research conducted by the research team using thematic analysis.

Did you perhaps also refer to the literature while conducting this analysis? If so, which theoretical body of work informed your assessment of the data? Do you have any citations to suggest evidence of how you enhanced the trustworthiness and validity, or relationship to theoretical grounding, in your work? Thank you we have corrected our method from co-production to workshop research. We have provided a detailed description of how we used literature to formulate our analysis process and coding framework now at the start of our results section. We believe the citations of the literature we provide here improve the trustworthiness and validity of our study and provide us with further theoretical grounding.

---

## [Decision Letter · Decision Letter 1]

11 Sep 2022

Developing a socio-ecological model for community engagement in a health programme in an underserved urban area

PONE-D-21-26119R1

Dear Dr. Caperon,

We’re pleased to inform you that your manuscript has been judged scientifically suitable for publication and will be formally accepted for publication once it meets all outstanding technical requirements.

Kind regards,

Ghaffar Ali, PhD

Academic Editor

PLOS ONE

Additional Editor Comments (optional):

Reviewers' comments:

Reviewer's Responses to Questions

**Comments to the Author**

1. If the authors have adequately addressed your comments raised in a previous round of review and you feel that this manuscript is now acceptable for publication, you may indicate that here to bypass the “Comments to the Author” section, enter your conflict of interest statement in the “Confidential to Editor” section, and submit your "Accept" recommendation.

Reviewer #1: All comments have been addressed

2. Is the manuscript technically sound, and do the data support the conclusions?

Reviewer #1: Yes

3. Has the statistical analysis been performed appropriately and rigorously? 

Reviewer #1: N/A

4. Have the authors made all data underlying the findings in their manuscript fully available?

Reviewer #1: Yes

5. Is the manuscript presented in an intelligible fashion and written in standard English?

Reviewer #1: Yes

6. Review Comments to the Author

Reviewer #1: As the revised manuscript satisfies the publication criteria, I approve of its publication.

The detailed explanation of the ethnic, economic, and cultural backgrounds of the study areas and participant communities are included in setting and the limitation section. Also, terminology modified to accepted and appropriate terminology for the study community.

In addition, I realized that a case study was used appropriately rather than a statistical analysis for the qualitative study. The literature reviews of the mentioned studies, sociological, and pertinent applied disciplines were beneficial for approval.

7. PLOS authors have the option to publish the peer review history of their article (what does this mean?). If published, this will include your full peer review and any attached files.

Reviewer #1: No

---

## [Editor Report · Acceptance letter]

13 Sep 2022

PONE-D-21-26119R1 

Developing a socio-ecological model for community engagement in a health programme in an underserved urban area 

Dear Dr. Caperon:

I'm pleased to inform you that your manuscript has been deemed suitable for publication in PLOS ONE. Congratulations! Your manuscript is now with our production department. 

Kind regards, 

on behalf of

Prof. Ghaffar Ali 

Academic Editor

PLOS ONE